rsob.royalsocietypublishing.org

**Subject Area:**
molecular biology/developmental biology

histone deacetylase, planthopper, fertility, courtship song

**Author for correspondence:**
Hai-Jun Xu
e-mail: haijunxu@zju.edu.cn

# The histone deacetylase NlHDAC1 regulates both female and male fertility in the brown planthopper, *Nilaparvata lugens*

Jin-Li Zhang, Xiao-Bo Yuan, Sun-Jie Chen, Hao-Hao Chen, Nan Xu, Wen-Hua Xue, Sheng-Jie Fu, Chuan-Xi Zhang and Hai-Jun Xu

State Key Laboratory of Rice and Ministry of Agriculture Key Laboratory of Molecular Biology of Crop Pathogens and Insect Pests, Institutes of Insect Sciences, Zhejiang University, Hangzhou 310058, People's Republic of China

C-XZ, 0000-0002-7784-1188; H-JX, 0000-0002-7314-377X

Histone acetylation is a specific type of chromatin modification that serves as a key regulatory mechanism for many cellular processes in mammals. However, little is known about its biological function in invertebrates. Here, we identified 12 members of histone deacetylases (NlHDACs) in the brown planthopper (BPH), *Nilaparvata lugens*. RNAi-mediated silencing assay showed that *NlHdac1*, *NlHdac3* and *NlHdac4* played critical roles in female fertility via regulating ovary maturation or ovipositor development. Silencing of *NlHdac1* substantially increased acetylation level of histones H3 and H4 in ovaries, indicating NlHDAC1 is the main histone deacetylase in ovaries of BPH. RNA sequencing (RNA-seq) analysis showed that knockdown of *NlHdac1* impaired ovary development via multiple signalling pathways including the TOR pathway. Acoustic recording showed that males with *NlHdac1* knockdown failed to make courtship songs, and thus were unacceptable to wild-type females, resulting in unfertilized eggs. Competition mating assay showed that wild-type females overwhelmingly preferred to mate with control males over *NlHdac1*-knockdown males. These findings improve our understanding of reproductive strategies controlled by HDACs in insects and provide a potential target for pest control.

## 1. Introduction

Histone acetylation is a specific type of chromatin modification that serves as a key regulatory mechanism for many cellular processes including DNA replication and regulation of gene expression [1–6]. The steady state of histone acetylation is determined by the antagonistic activities of histone acetyltransferases (HATs) and histone deacetylases (HDACs) [7]. The HATs acetylate the lysine residues on N-terminal tails of core histones, a process that generally correlates with gene activity [5]. Conversely, HDACs catalyse the removal of acetyl groups from lysine side chains on core histones and a range of other proteins [8], which is frequently associated with transcriptional repression. Since the discovery of the first *Hdac* [9], 18 mammalian genes encoding deacetylase activity have been identified [10]. Based on sequence similarity, these deacetylases are divided into two families, the classical HDAC family (zinc-dependent HDACs) and the sirtuin (sir2-like protein) family of NAD⁺-dependent deacetylases [11–15]. The classical HDAC family includes class I (HDACs 1, 2, 3 and 8), class II (HDACs 4, 5, 6, 7, 9 and 10) and class IV (HDAC11). Class III is represented by the sirtuin family which contains seven members (SIRTs 1–7), homologous to the yeast *Saccharomyces cerevisiae* Sir2 protein [16,17].

Of the 18 HDACs identified in mammals, HDAC1 is one of the most thoroughly studied at the biochemical and functional levels [18]. An early

rsob.royalsocietypublishing.org   Open Biol. 8: 180158

study revealed that germline deletion of *Hdac1* in mouse resulted in embryonic lethality due to severe proliferation defects and retardation in development [19]. Also, hundreds of studies on the growth-promoting activity of HDAC1 in human cancer were reported [20]. Notably, HDAC2 shares high amino acid sequence identity with HDAC1, and they have partially overlapping, but distinct roles in diverse biological processes [21]. For example, deleting of *Hdac2* only in oocytes led to subfertile mice whereas *Hdac1* and *Hdac2* double-mutant mice were infertile due to failure of DNA replication following fertilization [22,23]. Despite growing knowledge on the molecular mechanism of HDAC1 in mammals, little is known about its biological function in invertebrates. In the fruit fly *Drosophila melanogaster*, loss of *Rpd3*, the only *Hdac1* and *Hdac2* orthologue, led to a strong defect in embryo segmentation [24]. Additional studies suggested that Rpd3 was essential for cell survival, similar to the roles of HDAC1 and HDAC2 in mammalian cells [18,25–27]. Interestingly, a new biological function was assigned to Rpd3, which showed that wild-type flies subjected to a 7 h training session formed a robust long-term courtship memory, but this phenotype was completely abolished in the *Rpd3* mutant [28].

The migratory brown planthopper (BPH), *Nilaparvata lugens* (Hemiptera: Delphacidae), is a destructive pest of rice in most of Asia [29]. The BPH feeds exclusively on the phloem sap of the rice plant and can cause complete wilting and drying of plants, referred to as hopperburn [30,31]. In addition to direct feeding, BPH also transmits plant viruses such as rice ragged stunt virus and rice grassy stunt virus [32]. In past decades, this pest caused irregular but severe infestation throughout Asia, e.g. affecting an average annual area in 2005–2007 of about 26.7 million ha in China [33,34], thus leading to huge losses of rice yields. The BPH is characterized by r-strategy reproduction life history; therefore high fecundity is one of the most important biological features contributing to its ecological success [35]. In contrast to extensive studies of the effect of ecological factors on BPH fertility, the regulatory molecular mechanism has been little investigated. Recently, we sequenced and assembled the BPH genome [29], providing an avenue for better understanding of the molecular bases of BPH reproduction. Facilitated by a robust RNAi response [36,37], accumulated studies indicated that BPH fertility could be regulated epigenetically such as by DNA methylation [38], or at the transcriptional level [39–42]. This offers a potential strategy for developing RNAi-based pest control.

The aim of this study is to thoroughly investigate the biological functions of histone acetylation on BPH fertility by RNAi-mediated knockdown of the *NlHdac* family. First, we identified and characterized 12 *NlHdac* genes in BPH. Second, RNAi silencing assay showed that *NlHdac1*, *NlHdac3* and *NlHdac4* played pivotal roles in female fertility, and NlHDAC1 was the main HDAC in BPH ovary. Third, RNA-seq assay showed that NlHDAC1 regulated ovary development and oogenesis via multiple signalling pathways. Last, we showed that males with *NlHdac1* knockdown failed to make courtship songs and accomplish copulation, suggesting that *NlHdac1* plays an essential role in courtship and mating success of BPH males. Our results provide new insights into the role of HDACs in insects and offer a potential to develop NlHDAC1 inhibitors for BPH control.

# 2. Material and methods

## 2.1. Insects

The BPH was originally collected from a rice field in Hangzhou, China. Insects were maintained in a walk-in chamber at $26 \pm 0.5°C$ with a relative humidity of $50 \pm 5\%$ under a photoperiod of 16 L : 8 D. Insects were fed with rice seedlings (rice variety: Xiushui 134).

## 2.2. Identification and characterization of *NlHdac* genes in BPH

### 2.2.1. Gene identification and sequence analysis

The amino acid sequences of *D. melanogaster* HDACs were used to screen against *N. lugens* genomic and transcriptomic databases for identification of its homologues in BPH. Seventy HDAC sequences from 19 species including *Anopheles gambiae*, *Apis mellifera*, *Acyrthosiphon pisum*, *Bombyx mori*, *Bemisia tabaci*, *Cryptotermes secundus*, *Cyprinodon variegatus*, *D. melanogaster*, *Daphnia pulex*, *Eurytemora affinis*, *Gnatocerus cornutus*, *Halyomorpha halys*, *Neolamprologus brichardi*, *Notothenia coriiceps*, *Nasonia vitripennis*, *N. lugens*, *Stegastes partitus*, *Tribolium castaneum* and *Zootermopsis nevadensis* were included in the phylogenetic analysis. Conserved domains were predicted by SMART (http://smart.embl-heidelberg.de/). A phylogenetic tree was constructed using the MEGA5 program [43] with the method of maximum-likelihood and the bootstraps were set with 1000 replications.

### 2.2.2. Developmental profile and tissue distribution of HDACs

We used quantitative real-time PCR (qRT-PCR) to determine *NlHdac* and *NlSirt* transcripts in different developmental stages and tissues. For developmental profile examination, total RNAs were isolated from eggs ($n = 100$), first-instar ($n = 100$), second-instar ($n = 50$), third-instar ($n = 50$), fourth-instar ($n = 30$), fifth-instar nymphs ($n = 15$) and adult females ($n = 15$), which were laid or ecdysed within 24 h, using RNAiso Plus (Takara). For tissue distribution examination, we dissected head, gut, fat body, leg, cuticle and ovary from adult females ($n = 50$, 24 h after eclosion) for RNA extraction. Three independent biological replicates were set for RNA isolation. First-strand cDNA was synthesized using the PrimeScript 1st strand cDNA synthesis kit (Takara). The qRT-PCR was conducted on a CFX96TM real-time PCR detection system (Bio-Rad) with SYBR Supermix under the following conditions: denaturation for 3 min at 95°C, followed by 40 cycles at 95°C for 10 s and 60°C for 30 s. The specific primers corresponding to each *NlHdac* or *NlSirt* gene are listed in the electronic supplementary material, table S1. The relative expression levels of target genes were normalized by the *18S* rRNA gene using the $2^{-\triangle\triangle Ct}$ method (Ct represents the cycle threshold) [44]. Each sample was loaded for qRT-PCR with three technical replications.

## 2.3. *Nlhdac* knockdown and fecundity

### 2.3.1. RNAi and fecundity assay

The dsRNAs were synthesized using T7 high-yield transcription kit (Vazyme) according to the manufacturer's instructions with primers containing the T7 RNA polymerase

rsob.royalsocietypublishing.org    Open Biol. **8**: 180158

promoter at both ends (electronic supplementary material, table S1). The amplified sequences were verified by Sanger sequencing. The dsRNA injection was carried out as in our previous method [36,37]. Briefly, each fourth-instar individual was microinjected with approximately 150 ng of dsRNA. After injection, insects were maintained on fresh rice seedlings, which were renewed every 3 days.

Both virgin females and males at 3 days after adult eclosion were collected for fecundity analysis. Each female was allowed to match with two males in a glass tube. Insects were removed at designated time (5 or 10 days) and eggs counted under a stereomicroscope (Leica S8AP0).

### 2.3.2. Western blot analysis

Fourth-instar nymphs were treated with dsGfp, dsNlHdac1 or dsNlHdac4, and raised to adults. For immunoblot analysis of H3 and H4 acetylation, ovaries were dissected from females ($n = 50$) at $0-6$ h after eclosion, and total histones were extracted using the EpiQuik total histone extraction kit (Epigentek). For immunoblot analysis of vitellogenin, the fat body was dissected from females ($n = 50$) at 3 days after eclosion. Equal amounts of protein were loaded for each lane on SDS-PAGE gel. Western blotting was performed with antibodies of acetyl-histone H3 antibody sampler kit (Cell Signaling Technology), histone H3 acetylation antibody panel pack II (Epigentek), acetylhistone H4 antibody sampler kit (Cell Signaling Technology) and anti-vitellogenin mAb. Immuno-reactivity was imaged with the Molecular Imager ChemiDoc XRS+ system (Bio-Rad).

## 2.4. RNA-seq

### 2.4.1. Construction of cDNA libraries for Illumina sequencing

Fourth-instar nymphs ($0-24$ h interval after ecdysis) were injected with the dsRNA targeting either NlHdac1 or Gfp. At $0-6$ h after adult eclosion, ovaries were dissected either from dsNlHdac1- (dsHdac1_Ovary) or dsGfp-treated females (dsGfp_Ovary) for RNA preparation. Total RNA was isolated from 150 ovaries using RNAiso Plus (Takara) following the manufacturer's protocol. Experiments were performed in triplicate with three independently isolated RNA samples.

A total of 3 μg of RNA per sample was used as input material for RNA sample preparations. Sequencing libraries were generated using NEB Next Ultra™ RNA Library Prep Kit for Illumina (NEB) following the manufacturer's recommendations, and index codes were added to attribute sequences to each sample. In order to select cDNA fragments of $250 \sim 300$ bp in length, the library fragments were purified with AMPure XP system (Beckman) and library quality was assessed on the Agilent Bioanalyzer 2100 system. The clustering of the index-coded samples was performed on a cBot Cluster Generation System using TruSeq PE Cluster Kit v3-cBot-HS (Illumina) according to the manufacturer's instructions. After cluster generation, the library preparations were sequenced on the Illumina platform Hiseq X ten and 125/150 bp paired-end reads were generated.

### 2.4.2. Read mapping, normalization and quantification of expression differences

The clean reads were generated after removing adapter, ploy-N and low-quality reads from raw data. The clean reads were aligned to the reference BPH genome data (GCA_000757685.1_NilLug1.0) using Hisat2 (v. 2.0.4) [45]. HTSeq (v. 0.9.1) [46] was used to count the read numbers mapped to each gene. The number of fragments per kilobase of transcript sequence per million base pairs sequenced (FPKM) was used to calculate the gene expressions. The mapped reads of each sample were assembled by Cufflinks (v. 2.1.1) in a reference-based approach, and then the novel genes were predicted by Cufflinks (v. 2.1.1) [47].

Differential expression analysis between dsHdac1_Ovary and dsGfp_Ovary was performed using the DESeq R package (v. 1.18.0) [48]. Genes with an adjusted $p$-value < 0.05 found by DESeq were assigned as differentially expressed genes (DEGs). Gene Ontology (GO) enrichment analysis of DEGs was implemented by the GO seq R package [49] which could correct the gene length bias. The pathways enrichment of all DEGs was analysed in the KEGG pathway [50], and KOBAS software [51] was used to test the statistical enrichment of DEGs in KEGG pathways.

## 2.5. NlHdac knockdown and male fertility

### 2.5.1. Sperm viability assay

The sperm viability was determined using the Live/Dead sperm viability kit (Invitrogen) according to the manufacturer's instructions. Briefly, we firstly dissected testis from 30 individual males at 3 days after eclosion. Second, the semen was collected by cutting off the vas deferens, and subsequently pooled into 300 μl of buffer (10 mM HEPES, 150 mM NaCl, 10% BSA, pH 7.4). Third, SYBR 14 dye was added to semen solution, followed by incubation for 10 min at 36°C. After that, the solution was incubated with propidium iodide for another 10 min. Finally, the numbers of green (live) and red (dead) sperm in a 20 μl sample were counted using a Zeiss LSM 780 confocal microscope (Carl Zeiss MicroImaging). Three independent replications were performed.

### 2.5.2. Recording of courtship behaviour

Fourth-instar nymphs were injected with dsGfp or dsNlHdac1. For one pair-mating assay, one dsNlHdac1-treated male (3 days after eclosion) and one wild-type female (3 days after eclosion) were confined by a glass tube containing a single rice stem. The camera of an iPhone 6 cell phone was used to film the process of copulation from 20 insect pairs. Recording started when a female and a male were paired, and ended after 20 min. The dsGfp-treated males were allowed to match with wild-type females, which served as parallel controls. For the competitive mating assay, one dsNlHdac1-treated male, one dsGfp-treated male and one wild-type female were put together in one glass tube. Recording ended once the female finished copulation.

### 2.5.3. Detection of acoustic signals

The acoustic signals of BPH were sampled and analysed as in a previous report [52]. Briefly, fourth-instar nymphs were treated with dsRNAs targeting Gfp or NlHdac1. At 3 days after adult eclosion, one wild-type female was matched with either one dsGfp- or one dsNlHdac1-treated male. The acoustic signals were recorded with Adobe Audition, and

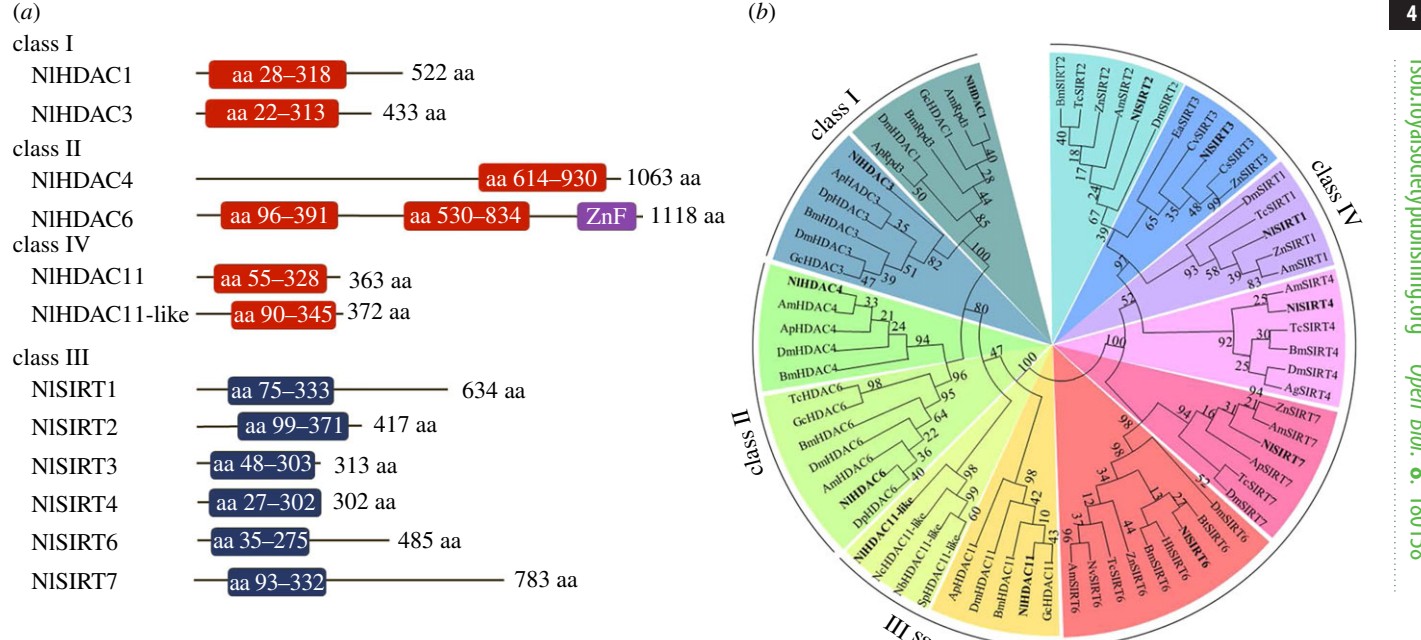

rsob.royalsocietypublishing.org   Open Biol. 8: 180158

**Figure 1.** Schematic depiction and phylogenetic analysis of the HDACs of BPH. (*a*) Schematic depiction of the HDAC members of BPH. The catalytic domain of deacetylase activity is shown by boxes in red or blue. The zinc finger domain of NlHDAC6 is shown by a purple box. The length of the open reading frame of each gene is shown on the right. (*b*) Phylogenetic analysis of the HDAC members of BPH. The phylogenetic tree was constructed based on 70 HDACs from 19 species (see Material and methods) using the maximum-likelihood method, and bootstraps were set with 1000 replications. The HDAC members of BPH are in bold.

MATLAB was used for data analysis. Recording of acoustic signals started once a female and a male were matched, and ended when the female finished copulation.

## 2.6. Image acquisition and data analysis

Images of insects and ovaries were captured with a DFC320 digital camera attached to a LEICA S8AP0 stereomicroscope using the digital imaging system LAS (v. 3.8). Statistical analysis was performed using SPSS (v. 20) and Microsoft EXCEL. Means were compared using two-tailed Student's *t*-test at the significance levels set at $*p < 0.05$ and $**p \leq 0.01$.

## 3. Results

### 3.1. Identification and characterization of *NlHdac* genes in BPH

#### 3.1.1. Identification of *NlHdac* genes in BPH

To comprehensively identify members of the HDAC family in BPH, we BLAST searched against its genomic [33] and transcriptome databases using the *D. melanogaster* HDAC proteins as query sequences. We identified 12 putative genes encoding HDACs, a number comparable to *Drosophila*, which has 10 members [53]. Out of 12 members in BPH, six shared similarity with the classic HDAC family of zinc-dependent deacetylases, designated NlHDAC1, NlHDAC3, NlHDAC4, NlHDAC6, NlHDAC11 and NlHDAC11_like (NlHDAC11_l); the remaining six members were close to the sirtuin family of NAD$^+$-dependent deacetylases, designated NlSIRT1–4, NlSIRT6 and NlSIRT7. The classical HDAC and the sirtuin families contained open reading frames ranging from 363 to 1118 and 302 to 783 amino acid

residues (figure 1*a*), respectively. Computer analysis revealed that each NlHDAC possessed one HDAC domain except for NlHDAC6 which had two HDAC domains and one zinc finger domain (figure 1*a*). Additionally, a catalytic core domain of the sirtuin family was predicted in NlSIRT1–7 (figure 1*a*). A phylogenetic analysis based on 70 HDAC orthologues from 19 species suggested that six NlHDACs in BPH were classified into three classes (figure 1*b*): NlHDAC1 and NlHDAC3 in class I, NlHDAC4 and NlHDAC6 in class II, and NlHDAC11 and NlHDAC11_l in class IV. The results also assigned NlSIRT1–7 together with their orthologues to class III (figure 1*b*), which formed an independent branch separated from the classical HDAC family. This phenomenon is consistent with the previous classification of the HDAC family. These events strongly indicated that the newly identified genes in BPH might encode proteins with HDAC enzymatic activity.

#### 3.1.2. Spatio-temporal expression patterns of HDACs in BPH

To better understand the function of *Hdac*, we first investigated their expressions across developmental stages using qRT-PCR. Our results revealed that all *NlHdac* and *NlSirt* genes except for *NlHdac1*, *NlHdac6*, *NlSirt4*, *NlSirt6* and *NlSirt7* had relatively stable expression levels throughout all life developmental stages (figure 2*a,b*). Higher levels of *NlHdac1*, *NlSirt6* and *NlSirt7* were expressed at the egg stage, indicating that they might play important functions in egg development; By contrast, *NlHdac6* and *NlSirt4* transcripts were abundant in nymph and adult stages, indicating that they might contribute more to organism growth.

Next, we determined tissue-specific expression profiles of the HDAC family in adult females. In general, most genes were widely expressed in all tissues examined, except for

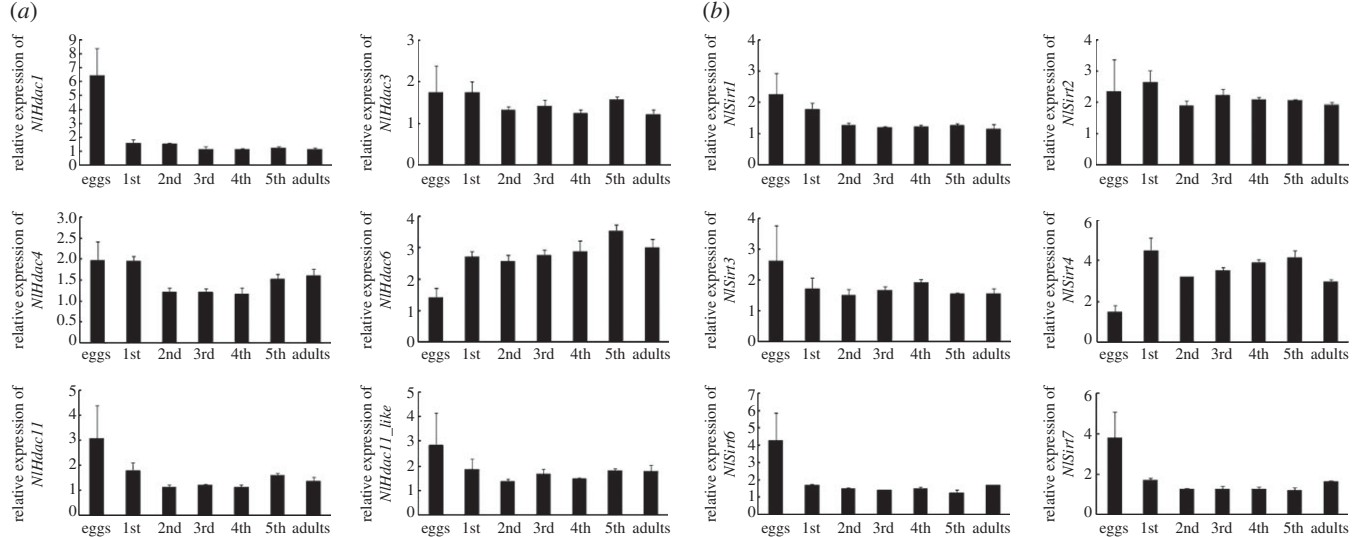

**Figure 2.** Developmental profile of the HDACs of BPH. (*a*) Developmental profile of the classical HDAC family (NIHDACs). (*b*) Developmental profile of the sirtuin family (NISIRTs). Total RNAs were isolated from eggs ($n = 100$), first-instar ($n = 100$), second-instar ($n = 50$), third-instar ($n = 50$), fourth-instar ($n = 30$), fifth-instar nymphsinstar ($n = 15$) and adult females ($n = 15$), which were laid or ecdysed within 24 h. First-strand cDNA was synthesized using random primers, and qRT-PCR was conducted using specific primers corresponding to each gene. The relative expression level was normalized by the *18S* rRNA gene. Bars represent s.e.m. derived from three independent biological replicates.

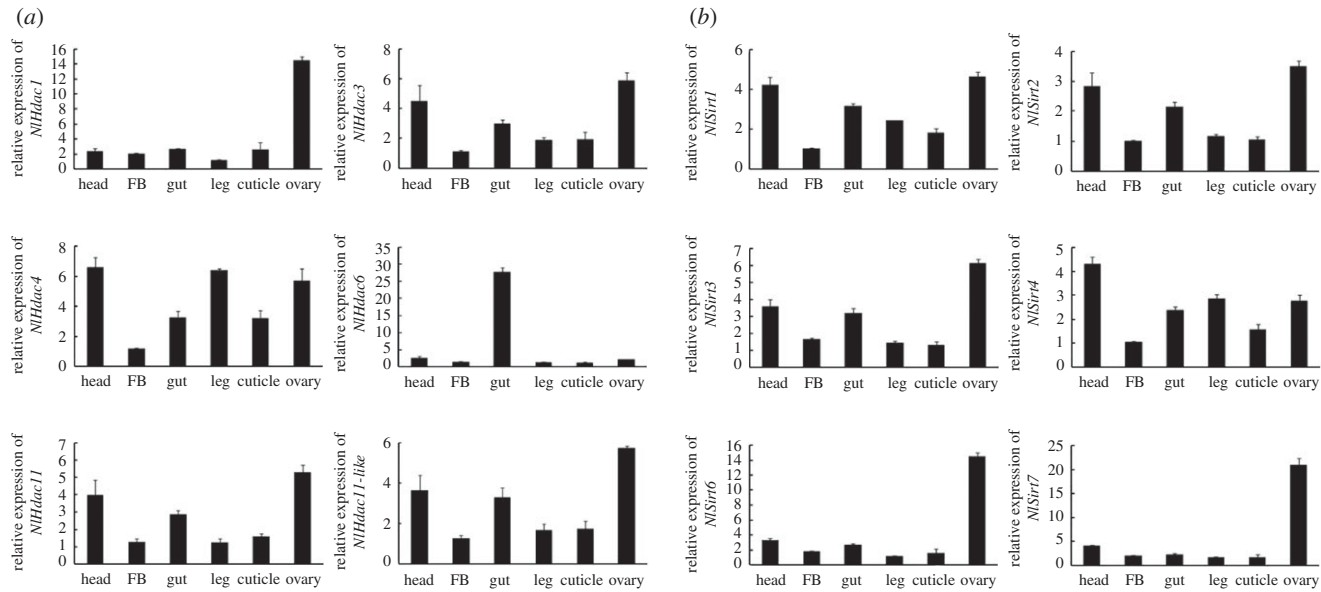

**Figure 3.** Tissue distribution of the HDACs of BPH. (*a*) Tissue distribution of the classical HDAC family (NIHDACs). (*b*) Tissue distribution of the sirtuin family (NISIRTs). Total RNAs were isolated from head, gut, fat body (FB), leg, cuticle and ovary from adult females ($n = 50$, 24 h after eclosion). First-strand cDNA was synthesized using random primers, and qRT-PCR was conducted using specific primers corresponding to each gene. The relative expression level was normalized by the *18S* rRNA gene. Bars represent s.e.m. derived from three independent biological replicates.

*NlHdac1*, *NlHdac6*, *NlSirt6* and *NlSirt7* (figure 3*a,b*). The *NlHdac6* almost solely occurred in the gut tissue (figure 3*a*); however, *NlHdac1*, *NlSirt6* and *NlSirt7* had considerably higher levels in the ovary relative to other tissues, suggesting they were involved in ovary development and oogenesis.

## 3.2. *NlHdac* knockdown and female fecundity

### 3.2.1. Knockdown of *NlHdac1*, *NlHdac3* or *NlHdac4* leads to female infertility

To investigate whether the zinc-dependent HDACs play roles in BPH fertility, fourth-instar nymphs were challenged with corresponding dsRNAs targeting each classical *NlHdac*

gene. The qRT-PCR analysis showed that microinjection of dsRNA significantly reduced expression of each gene relative to the ds*Gfp* treatment (electronic supplementary material, figure S1). After adult eclosion, females were allowed to mate with males to deposit eggs for 10 days. Both ds*NlHdac1*- and ds*NlHdac4*-treated females produced very few eggs (figure 4*a*), in contrast to more than 200 eggs produced by ds*Gfp*-treated females, which served as a parallel control. Because most ds*NlHdac3*-treated females failed to survive for 10 days during adulthood, we counted the amount of eggs laid within the first 5 days after adult eclosion. Similar to ds*NlHdac1*- and ds*NlHdac4*-treated females, ds*NlHdac3*-treated individuals were nearly infertile (figure 4*b*). In addition, knockdown of *NlHdac6*,

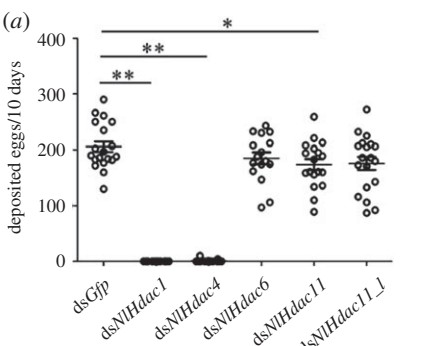
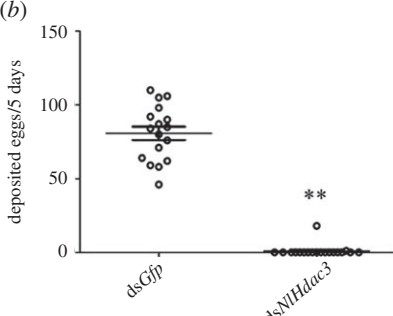
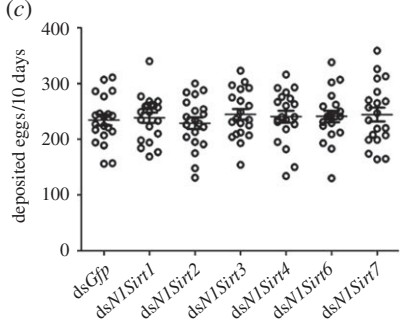

**Figure 4.** Fecundity of females with gene knockdown. (a) Number of eggs produced by females treated with ds*Gfp* (n = 19), ds*NlHdac1* (n = 19), ds*NlHdac4* (n = 19), ds*NlHdac6* (n = 16), ds*NlHdac11* (n = 18) or ds*NlHdac11_like* (ds*NlHdac11_l*) (n = 19). (b) Number of eggs produced by females treated with either ds*Gfp* (n = 17), or ds*NlHdac3* (n = 19). (c) Number of eggs produced by females treated with either ds*Gfp* (n = 20), ds*NlSirt1* (n = 20), ds*NlSirt2* (n = 20), ds*NlSirt3* (n = 20), ds*NlSirt4* (n = 20), ds*NlSirt6* (n = 20) or ds*NlSirt7* (n = 20). Females were allowed to deposit eggs for 5 days in (b), and for 10 days in (a) and (c). Each circle represents eggs produced by an individual female. Bar represents mean ± s.e.m. Statistical comparisons between two groups were performed using two-tailed Student's *t*-test (*$p < 0.05$; **$p < 0.01$).

rsob.royalsocietypublishing.org Open Biol. 8: 180158

*NlHdac11* or *NlHdac11_l* slightly decreased BPH fecundity (figure 4a), among which the result for ds*NlHdac11* was significant compared to ds*Gfp* treatment. These observations indicate that the zinc-dependent HDACs were involved in BPH fecundity, with NlHDAC1, NlHDAC3 and NlHDAC4 playing central roles.

Next, we investigated whether the NAD⁺-dependent HDACs (class III) affected female fecundity. Similar amounts of eggs were produced by females treated with ds*NlSirt1*, ds*NlSirt2*, ds*NlSirt3*, ds*NlSirt4*, ds*NlSirt6* or ds*NlSirt7* compared with ds*Gfp* treatment (figure 4c), implying that these HDACs played marginal roles in BPH fertility.

### 3.2.2. Knockdown of *NlHdac1*, *NlHdac3* or *NlHdac4* leads to undeveloped ovaries or malformed ovipositor

Because severe defects in egg deposition were derived from ds*NlHdac1*, ds*NlHdac3* and ds*NlHdac4*, we subsequently focused on these three genes to better understand the underlying mechanism. First, we examined external reproductive organs of females after dsRNA treatment. We noted that first valvifers of ds*NlHdac1*- or ds*NlHdac3*-treated females extended more widely than those of ds*Gfp*-treated females (figure 5). Closer inspection showed that knockdown of *NlHdac1*- or *NlHdac3* led to a loosely organized ovipositor, which is usually closely compact with ventral, inner and dorsal valvulae as seen in ds*Gfp*-treated females. To successfully deposit eggs into a rice stem, BPH has to penetrate through rigid rice sheaths with its ovipositor, and so malformed ovipositors might severely impair egg deposition. For ds*NlHdac4*-treated individuals, there was no discernible difference in morphology (figure 5). Second, we examined ovary development in females at 3-day- and 5-day-adulthood. For ds*Gfp*-treated females, ovaries were fully developed, and each ovary tube was filled with a banana-like oocyte (figure 5). By contrast, ovaries of ds*NlHdac1*- or ds*NlHdac4*-treated 3-day-females were small and poorly developed, and remained immature up to 5 days (figure 5), indicating impaired ovary maturation and oogenesis. Unexpectedly, ds*NlHdac3*-treated females had normal ovaries similar to those of ds*Gfp*-treated females, albeit the former were infertile (figure 5). Consequently, we ascribed infertility in ds*NlHdac3*-treated females and ds*NlHdac1*- or ds*NlHdac4*-treated females to ovipositor malformation and immature ovaries, respectively.

### 3.2.3. NlHDAC1 is a major deacetylase in BPH ovary

Because HDAC can catalyse the removal of acetyl groups from lysine side chains on core histones, knockdown of *Hdac* would hypothetically increase acetylation levels of certain lysine sites on histones (hyperacetylation). To gain further insight into the regularity of ovary development in ds*NlHdac1*- or ds*NlHdac4*-treated females, we surveyed the acetylation status of histones H3 and H4 by western blot analysis using commercially available anti-acetyl antibodies. The results showed that knockdown of *NlHdac1* not only substantially increased acetylation levels of five lysine sites on histone H3 (H3K9, H3K14, H3K18, H3K23 andH3K27), but also four lysine sites on histone H4 (H4K5, H4K8, H4K12 and H4K16), compared to the basal level of histone acetylation in ds*Gfp*-treated ovaries (figure 6). This result suggests that NlHDAC1 widely deacetylases histones rather than targeting specific lysines on histones in the ovary. In a concurrent trial, no change of acetylation level was observed in ds*NlHdac4*-treated ovaries compared to ds*Gfp* treatment (figure 6). The above evidence indicates that NlHDAC1 was a major deacetylase in the BPH ovary, and played a vital role in ovary maturation.

In insects, vitellogenin is the main resource for sustenance of developing eggs [54,55], and silencing either of *vitellogenin* or its receptor gene phenocopied the ds*NlHdac1*- or ds*NlHdac4*-treatment, leading to arrested development of ovaries [56]. To investigate whether vitellogenin contributes to the defects of ovary development in ds*NlHdac1*- and ds*NlHdac4*-treated females, we then determined the *vitellogenin* expression in the fat body, the main organ producing vitellogenin in insects. Western blot analysis demonstrated that knockdown of either *NlHdac1* or *NlHdac4* produced similar amounts of vitellogenin compared to the ds*Gfp* treatment (electronic supplementary material, figure S2). This indicated that NlHDAC1 and NlHDAC4 regulated ovary development through unknown factors, not through vitellogenin.

rsob.royalsocietypublishing.org  Open Biol. 8: 180158

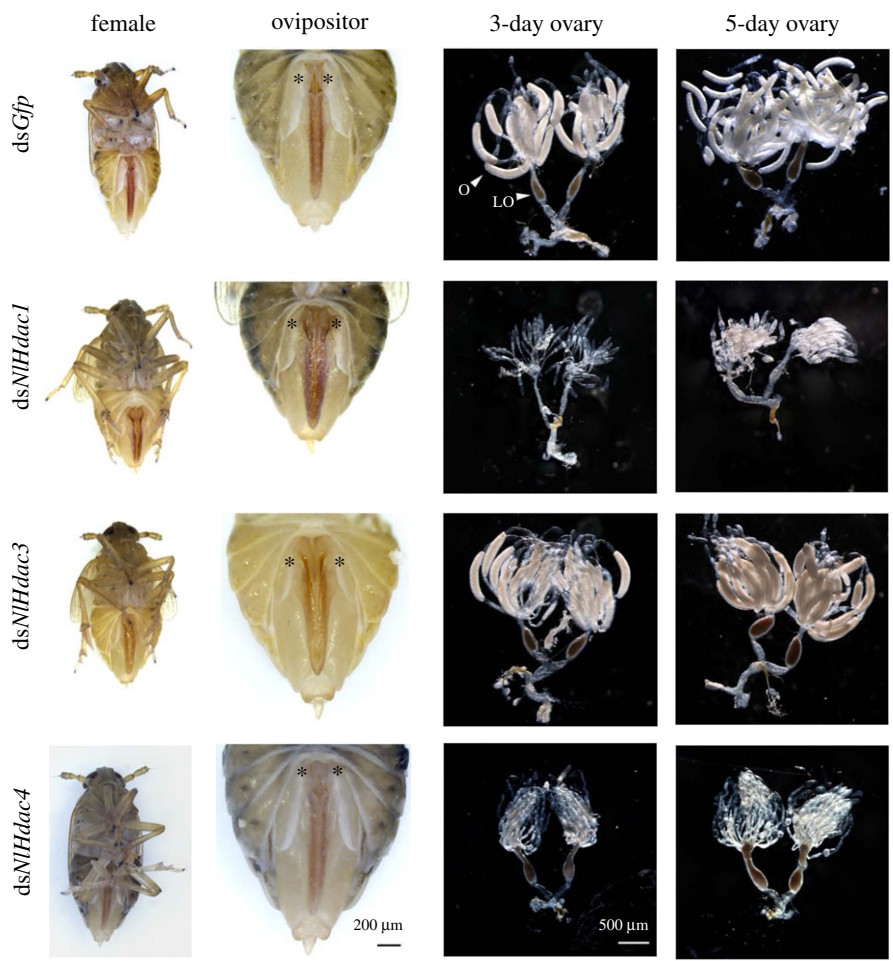

**Figure 5.** Morphologies of ovipositor and ovary upon gene knockdown. Fourth-instar nymphs were microinjected with dsRNA targeting *Gfp*, *NlHdac1*, *NlHdac3* or *NlHdac4*. Ovaries were dissected from females at 3 or 5 days after adult eclosion. The first valvifers of females are shown by stars. Ovary (O) and lateral oviduct (LO) are indicated by arrow heads.

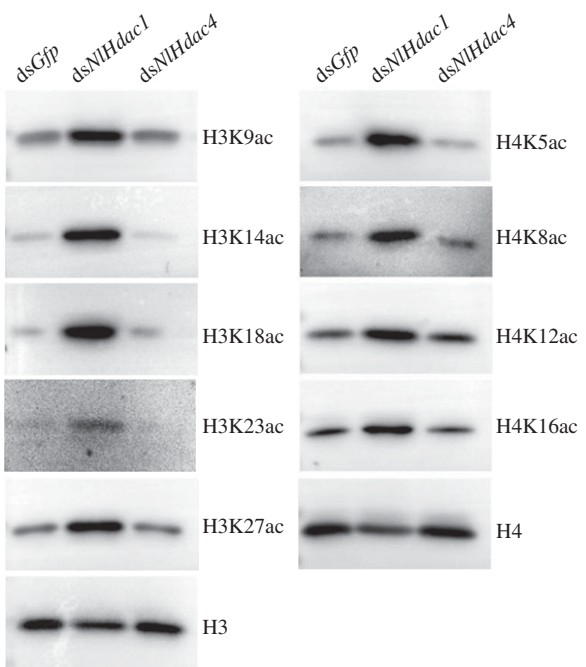

**Figure 6.** Acetylation levels of histone H3 and H4 in ovary. Fourth-instar nymphs were treated with dsRNA targeting *Gfp*, *NlHdac1* or *NlHdac4*. Ovaries dissected from females (*n* = 50) at 0–6 h after eclosion were used for western blot analysis. Monoclonal antibodies specific to acetylated lysines on histones are indicated on the right. Total amounts of H3 and H4 histones were used for loading control.

## 3.3. NlHDAC1 regulates ovary maturation via multiple signalling pathways

Since chromatin modification transcriptionally changes gene expression, we collected ovaries from females treated with either ds*NlHdac1* or ds*Gfp* for RNA-seq (electronic supplementary material, file S1). Our results revealed that 5725 (25.5%) genes were differentially expressed genes (DEGs) with adjusted *p*-value < 0.05, among which 3180 (14.2%) genes had higher and 2545 (11.3%) had lower expression in ovaries treated with ds*NlHdac1*, compared with ds*Gfp* (figure 7*a*; electronic supplementary material, additional file S1). Further analysis demonstrated that these DEGs were mapped to 116 Kyoto Encyclopedia of Genes and Genomes (KEGG) pathways including the mTOR, MAPK, Notch, Hippo and Wnt signalling pathways associated with cell growth, proliferation and differentiation (figure 7*b*). The mTOR signalling pathway received our attention because a previous report showed that knockdown of *NlTor* in BPH females arrested ovary development and oogenesis [39]. Accordingly, we first verified the downregulation of *NlTor* in ds*NlHdac1*-treated ovaries by qRT-PCR (electronic supplementary material, figure S3). Subsequently, we knocked down *NlTor* in BPH females, which led to undeveloped ovaries compared to ds*Gfp*-treatment (figure 7*c*). This phenomenon indicated that NlHDAC1 regulated ovary maturation and oogenesis through multiple signalling pathways including the mTOR pathway.

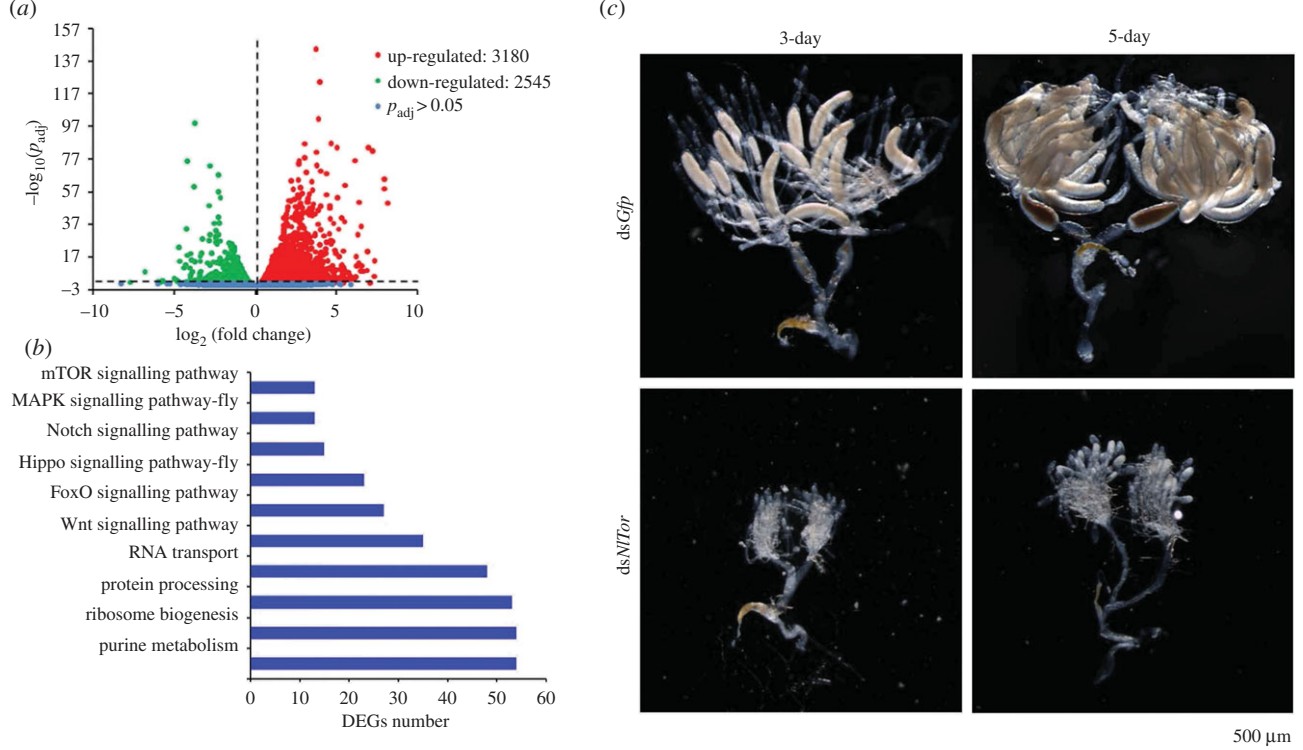

**Figure 7.** Differentially expressed genes (DEGs) in ovary upon *NlHdac1* knockdown. (*a*) Numbers of upregulated and downregulated genes in ds*NlHdac1*-treated ovaries compared to the ds*Gfp* treatment. (*b*) Selected KEGG pathways derived from DEGs (10 of 116 enriched KEGG pathways). (*c*) Phenotype of ovaries upon *NlTor* knockdown. Ovaries treated with ds*Gfp* served as the parallel control.

## 3.4. *NlHdac* knockdown and male fertility

### 3.4.1. Knockdown of *NlHdac1* impairs fertility of BPH males

Given that knockdown of *NlHdac1*, *NlHdac3* and *NlHdac4* led to female infertility, it was possible that they also affected male fertility. Accordingly, males treated with ds*NlHdac1*, ds*NlHac3* or ds*NlHdac4* were allowed to mate with wild-type females. Subsequently, eggs deposited in 10 days were collected and analysed. Knockdown of either *NlHdac3* or *NlHdac4* resulted in a similar amount of eggs compared with ds*Gfp* treatment (figure 8*a*). However, wild-type females that mated with ds*NlHdac1*-treated males deposited significantly fewer eggs (figure 8*a*). Furthermore, the ds*NlHdac1*-treated eggs failed to accumulate eye pigmentation, a characteristic hallmark of egg development, even up to 6 days after egg deposition (figure 8*b*; electronic supplementary material, figure S4). This observation was reminiscent of unfertilized eggs, which showed no eye pigmentation throughout the whole egg stage (figure 8*c*). Conversely, ds*NlHdac3*- and ds*NlHdac4*-treated eggs exhibited the same eye pigmentation as ds*Gfp*-treated eggs (figure 8*b*), suggesting that these eggs were fertilized. In all, these observations indicate that *NlHdac1* but not *NlHdac3* or *NlHdac4* played a critical role in male fertility.

Next, we examined testis development and sperm viability in ds*NlHdac1*-treated males. The ds*NlHdac1*-treated males had moderately smaller accessory glands than ds*Gfp*-treated males (figure 9*a*). However, there was no significant difference in sperm vitality between *dsNlHdac1*- and *dsGfp*-treated males (figure 9*b*,*c*), indicating that infertility of *dsNlHdac1*-treated males was due to factors other than sperm vitality.

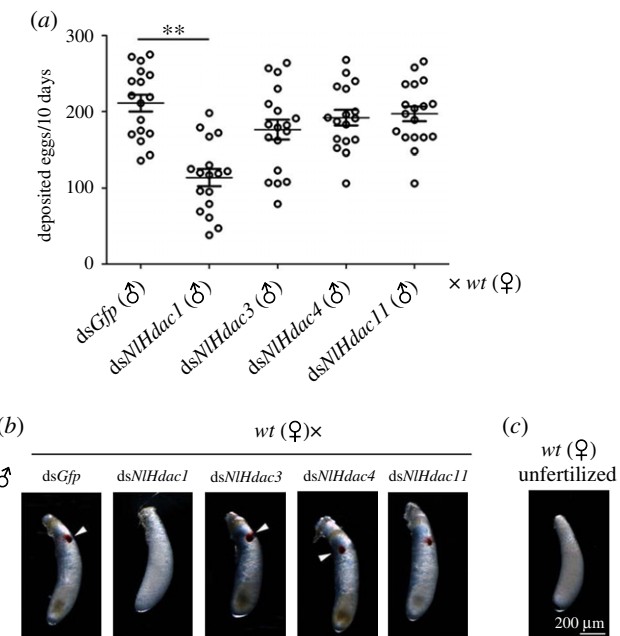

**Figure 8.** Fecundity of males with gene knockdown and phenotype of eggs produced. Fourth-instar nymphs were treated with dsRNAs targeting each designed gene. At 3 days after adult eclosion, dsRNA-treated males were allowed to mate with wild-type females. (*a*) Number of eggs produced in paired mate assay. Wild-type females were mated with males that were previously treated with either ds*Gfp* (*n* = 17), ds*NlHdac1* (*n* = 17), ds*NlHdac3* (*n* = 18), ds*NlHdac4* (*n* = 17) or ds*NlHdac11* (*n* = 18). Each circle represents eggs produced by an individual female. Bar represents mean ± s.e.m. Statistical comparisons between two groups were performed using two-tailed Student's *t*-test (**$p < 0.01$). (*b*) Morphology of eggs deposited in paired mate assay. The eye pigmentation is indicated by arrow heads. (*c*) An unfertilized egg was deposited by a virgin wild-type female, which showed no eye pigmentation across the egg stage, and never hatched. *wt*, wild-type.

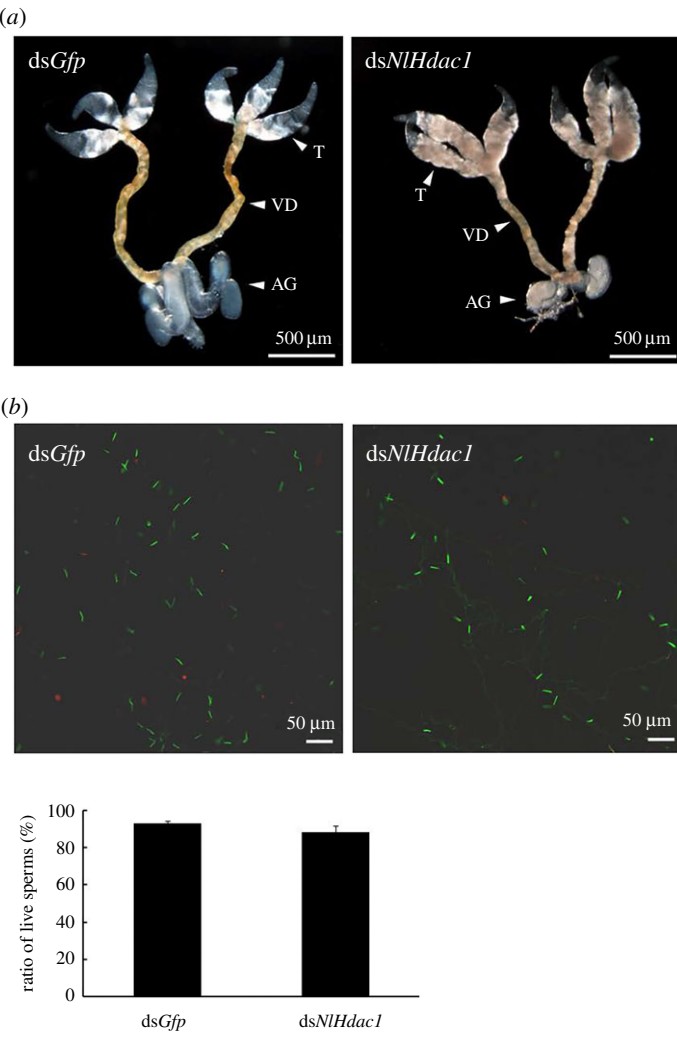

**Figure 9.** Testis development and sperm viability in ds*NlHdac1*-treated males. Fourth-instar nymphs were treated either with ds*Gfp* or ds*NlHdac1*, and testis was dissected from males ($n = 20$) at 3 days after eclosion (*a*). (*b*) The sperm viability in ds*Gfp*- or ds*NlHdac1*-treated males. Live and dead sperms were shown in green and red, respectively. Testis (T), vas deferens (VD) and accessory gland (AG) were indicated by arrow heads.

### 3.4.2. Wild-type females reject copulation with ds*NlHdac1*-treated males

Because wild-type females that mated with ds*NlHdac1*-treated males produced unfertilized-like eggs, and because mating is a prerequisite for fertilized eggs, we performed a paired-mating assay using ds*NlHdac1*-treated males and wild-type females. The courtship of normal BPH males consists of a series of behaviours prior to copulation, including orientation toward the female, abdomen vibration for courtship song generation, tapping and attempted copulation [57] (also see figure 10). In the paired mating assay ($n = 20$), the ds*NlHdac1*-treated males actively approached the wild-type female and subsequently attempted copulation soon after the virgin wild-type female emitted the acoustic signal by vibrating her abdomen (electronic supplementary material, movie S1). However, the wild-type female vigorously rejected copulation either by walking away when the male approached, or by kicking toward the male prior to genital contact, or by failing to raise her abdomen (electronic supplementary material, movie S1). In the parallel experiment, the ds*Gfp*-treated males ($n = 20$) successfully copulated with the wild-type female as expected (electronic supplementary material, movie S2). It took about 200 s for ds*Gfp*-treated males to finish copulation after female

vibration (electronic supplementary material, figure S6a), and copulation duration was approximately 100 s (electronic supplementary material, figure S6b). To confirm the above observation, we next performed a mate choice assay ($n = 20$). When presented simultaneously with a choice of ds*Gfp*- and ds*NlHdac1*-treated males, the wild-type female 100% preferred mating with the ds*Gfp*-treated male (electronic supplementary material, movie S3). These observations indicated that *NlHdac1*-knockdown somehow impaired male courtship, thus leading to failed copulation.

### 3.4.3. Males with *NlHdac1* knockdown cannot make courtship songs

Given that both male and female BPHs emit acoustic signals prior to mating, which serve as an important cue used by the female to recognize its mate [58], we recorded acoustic signals during the single-pair mating assay. The acoustic signals are produced through abdominal vibration by males and females and are transmitted through rice plants [59–61]. The pattern of acoustic signals (courtship songs) from the wild-type male consists of several rapid pulses (chirp song) and heavily damped pulses (buzz song) (figure 11*a*), whereas courtship from the wild-type female consists of rhythmic repeated pulses (figure 11*b*). For a

rsob.royalsocietypublishing.org    Open Biol. **8**: 180158

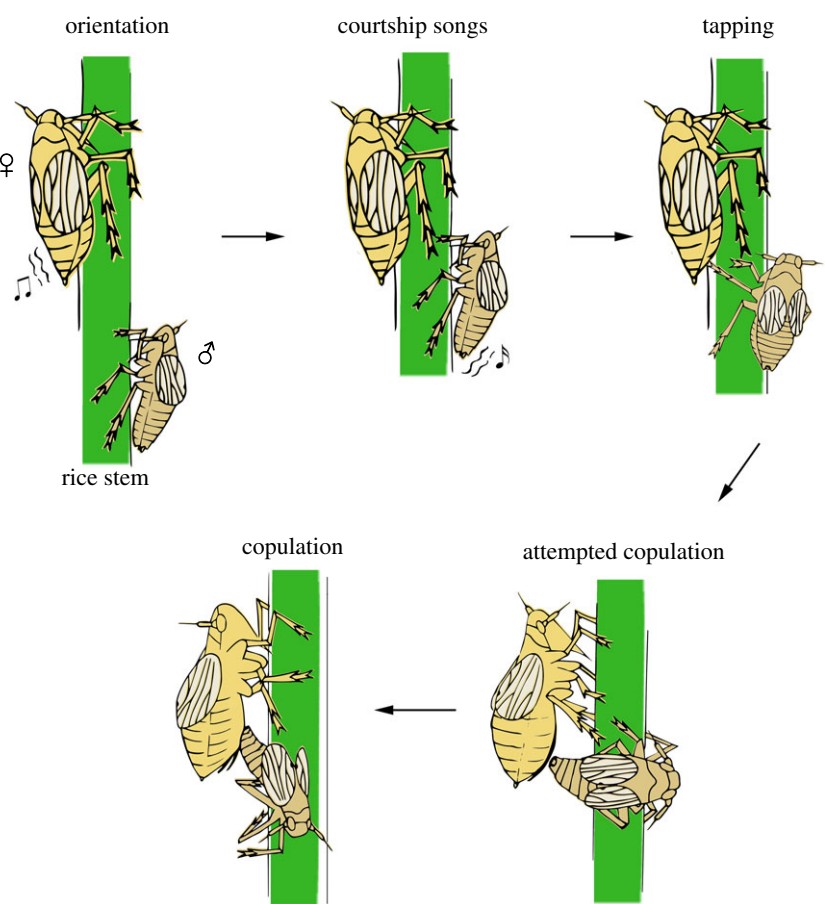

**Figure 10.** Schematic depiction of the courtship of wild-type BPH males. The process of courtship includes a series of steps: (1) the male orientates towards the female once the latter emits acoustic signals, (2) the male vibrates its abdomen for courtship song generation, (3) the male taps the abdomen of the female with its fore legs or middle legs, (4) the male attempts copulation with the female and (5) successful copulation occurs.

single pair of ds*NlHdac1*-treated male and wild-type female ($n = 20$), courtship songs only from the female but not the male were recorded (figure 11*d*). In the parallel control experiment ($n = 20$), we readily detected courtship songs from both wild-type females and ds*Gfp*-treated males (figure 11*c*). These events indicated that dysfunction of NlHDAC1 impaired the ability of BPH males to make courtship songs, thus leading to failed copulation, and is possibly the main reason for infertility of ds*NlHdac1*-treated males.

# 4. Discussion

In this study, we identified 12 members of the HDAC family in BPH, including six members of the classical HDAC family (NlHDACs) and the remaining six members of the sirtuin family of NAD$^+$-dependent deacetylases (NlSIRTs). The RNAi-based functional analysis indicated that all the *NlHdac*s were more or less involved in female fecundity, with *NlHdac1*, *NlHdac3* and *NlHdac4* playing the most important roles. Notably, knockdown of *NlHdac1*, *NlHdac3* or *NlHdac4* led to female infertility (figure 4*a*,*b*). This effect was consistent with the egg-stage dependent (figure 2*a*) and ovary-biased expression pattern of *NlHdac1* (figure 3*a*). By contrast, *NlSirt*s had no effect on female fecundity via RNAi silencing analysis (figure 4*c*). More evidence showed that the infertility of ds*NlHdac1*- or ds*Hdac4*-treated females was most probably due to undeveloped ovaries (figure 5), but dysfunction of *NlHdac3* led to a malformed ovipositor that might render females unable to deposit eggs inside the rice stem (figure 5).

## 4.1. NlHDAC1 is the main histone deacetylase in ovaries of BPH

In attempting to establish a functional link between acetylated lysines and undeveloped ovaries caused by *NlHdac1*- or *NlHdac4*-knockdown, we examined the acetylation level in a subset of histones H3 and H4 in ovaries with available anti-acetylation antibodies. Western blot analysis showed that knockdown of *NlHdac1* led to hyperacetylation of all lysine side chains on histones instead of targeting a specific one (figure 6), indicating that NlHDAC1 was a major deacetylase in the BPH ovary. This finding is consistent with previous observations for *Drosophila* and mouse cell lines. Silencing of *Hdac1* in *Drosophila* S2 cells, but not of the other *Hdac* family members, led to increased histone acetylation [25]. In embryonic stem cells of mouse, *Hdac1* null led to modest hyperacetylation of histones H3 and H4, suggesting that HDAC1 was a major deacetylase in these cells [62]. Unfortunately, the general hyperacetylation pattern of ds*NlHdac1* prevented us further identifying specific lysine residues that were responsible for regulating ovary development in BPH. In addition, our parallel experiments showed that ovaries harbouring *NlHdac4*-knockdown had similar levels of histone acetylation to the ds*Gfp* treatment (figure 6), indicating that NlHDAC4 played a marginal role in histone acetylation in ovaries. Alternatively, NlHDAC4 could deacetylate an additional subset of lysine residues, which were neglected by us in this study. Hence, the detailed

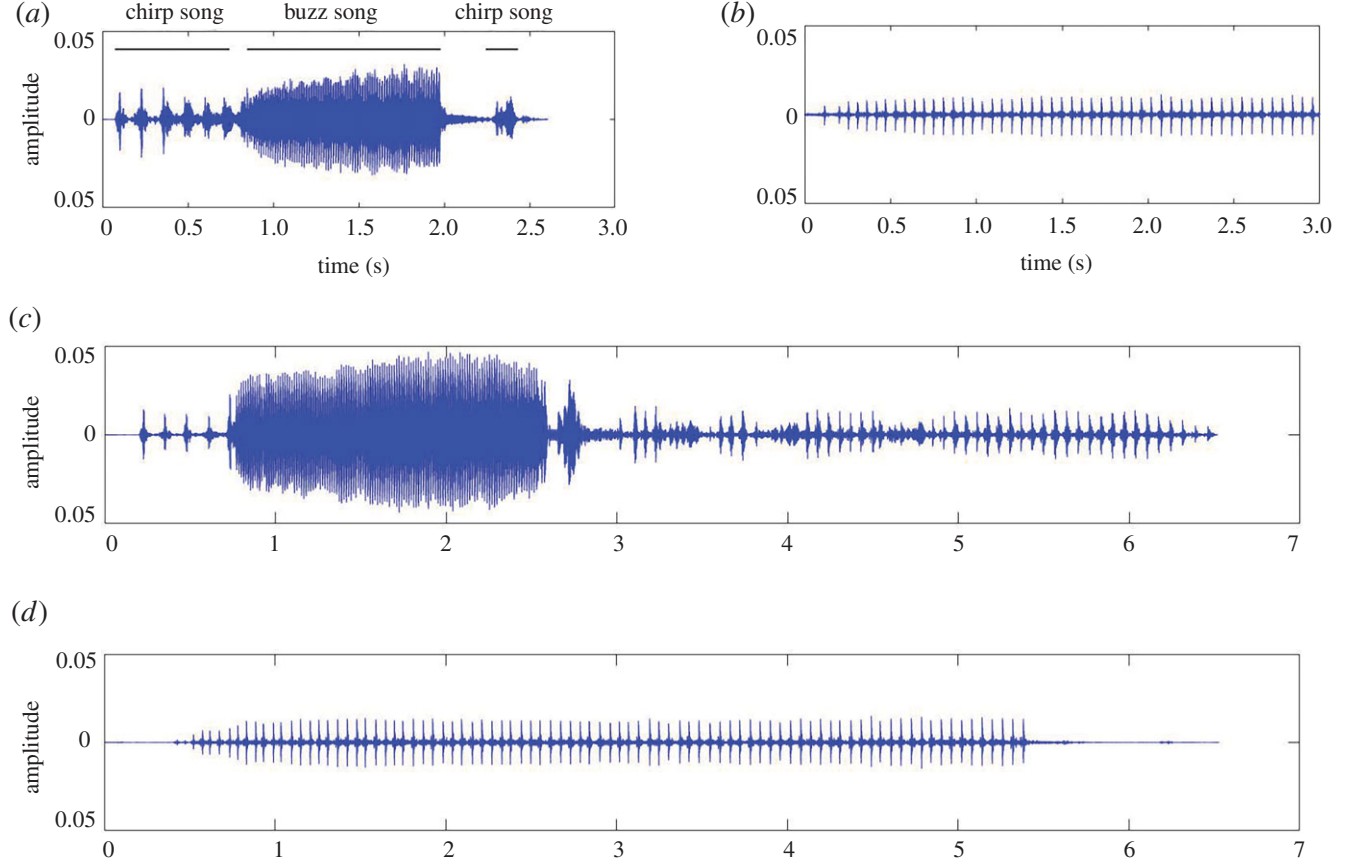

**Figure 11.** Oscillograms of female and male courtship songs of BPH. (*a*) Oscillogram of courtship songs of wild-type males, consisting of chirp songs and buzz songs. (*b*) Oscillogram of courtship songs of wild-type females. (*c*) Courtship songs recorded from paired mating assay (*n* = 20) which contained one ds*Gfp*-treated male and one wild-type female. (*d*) Courtship songs recorded from paired mating assay (*n* = 20) which contained one ds*NlHdac1*-treated male and one wild-type female.

rsob.royalsocietypublishing.org *Open Biol.* **8**: 180158

## 4.2. Transcriptomic analysis in ovaries with *NlHDAC1* knockdown

Contributing to deacetylase activity, HDACs are widely believed to remove the acetyl groups from lysine side chains on histones and thereby favour transcriptional repression through chromatin compaction [8,63]. Given that NlHDAC1 was found to be a major deacetylase in BPH ovaries, we investigated the genome-wide transcriptional change by RNA-seq after gene knockdown (electronic supplementary material, Additional file S1). Compared to the ds*Gfp*-treated ovary, there were 5725 (25.5%) DEGs after *NlHdac1* knockdown. Among them, 3180 were significantly upregulated, giving strong support to the conventional notion of HDACs as transcription repressors [64]. However, we also found that knockdown of *NlHdac1* downregulated expression of 2545 genes, a similar number to that of upregulated genes. A similar phenomenon was also observed in *Hdac1*-null embryonic stem cells of mouse, in which 4% of genes were upregulated and 3% were downregulated [8,62]. Additionally, *Drosophila* treated with the HDAC inhibitor 4-phenylbutyrate resulted in upregulated or downregulated expression of hundreds of genes [65]. These data strongly

suggest that NlHDAC1 serves as both a transcriptional repressor and an activator in BPH ovaries.

Among the 5725 DEGs, the category of metabolic pathways was the most enriched (373 genes) in KEGG analysis, followed by several well-known signalling pathways including mTOR, MAPK, Wnt and Hippo signalling pathways (electronic supplementary material, table S6). These data indicated that NlHDAC1 might regulate gene expression via cross-talking with multiple signalling pathways. The mTOR signalling pathway coordinates cell growth with environmental conditions and plays a fundamental role in cell and organismal physiology [66]. The MAPK pathways sense aspects of the extracellular environment and regulate a variety of cellular processes, including proliferation and differentiation [67]. The Wnt signalling pathway is important for stem cell renewal, cell proliferation and cell differentiation both during embryogenesis and adult tissue homeostasis [68,69]. The Hippo pathway is a universal governor of organ size, tissue homeostasis and regeneration, which regulates cell proliferation, differentiation and spatial patterning in organ development [70,71]. Among them, the mTOR signalling pathway received particular attention because a previous report showed that *NlTor*-knockdown in BPH females arrested ovary development and oogenesis [39]. Hansen *et al*. [72] reported that knockdown of *Tor* inhibited vitellogenesis in the case of the mosquito *Aedes aegypti*, thus inhibiting egg development. In this study, we verified that *NlHdac1*-knockdown reduced *NlTor* transcripts by qRT-PCR

(electronic supplementary material, figure S3), and observed undeveloped ovaries in ds*NlHdac1*-treated females (figure 7*c*). Taken together, it is plausible that NlHDAC1 regulated ovary maturation and oogenesis in part through the mTOR signalling pathway. Unfortunately, we were unable to test this proposal via a rescue experiment because of the lack of genetic tools to investigate activation of the mTOR pathway in females with *NlHdac1*-knockdown. In addition, given that NlHDAC1 serves as an overall regulator of chromatin modification, it is difficult to determine a particular pathway that is specifically involved in ovary maturation and oogenesis.

### 4.3. ds*NlHdac1*-treated males failed to make courtship songs

Courtship songs have evolved in association with mate recognition and mate preference, and serve as attractive signals in most animals [73]. Notably, *NlHdac1*-knockdown males failed to make courtship songs (figure 11). As a result, wild-type females rejected mating with ds*NlHDAC1*-treated males in the paired mate assay (electronic supplementary material, movie S1), overwhelmingly preferring ds*Gfp*- over ds*NlHdac1*-treated males in the mate competition assay (electronic supplementary material, movie S3). These results demonstrate that NlHDAC1 played an essential role in courtship and mating success of BPH males. At present, there is little knowledge of the underlying mechanism by which NlHDAC1 affects courtship song. In *Drosophila*, courtship song is specified by a fairly small set of neurons that express a duet of sex hierarchy genes, *doublesex* and *fruitless* [74–76]. An additional study indicated that HDAC1 regulated the establishment of a sexually dimorphic single-neuron by interacting with the Bonus and Fruitless complex, and then affected courtship in *Drosophila* [77]. Based on these studies, it will be of interest to determine whether NlHDAC1 regulates courtship songs through affecting Fruitless-expressing neurons in BPH.

The migratory BPH feeds exclusively on rice, and, in East Asia, only overwinter in tropical or subtropical areas (Vietnam and southern China). In spring and summer, they migrate northward as rice becomes available in temperate areas of China, Japan and Korea [78,79], thus causing massive outbreaks. Flight is energetically costly, and fitness trade-offs

between flight capability and life-history traits exist. Migratory individuals prolonged the age at first reproduction, and produced fewer eggs than sedentary individuals [80]. As the most abundant post-translational modifications, whether the reversible acetylation of histones and HDACs contribute to this process warrants further investigation.

## 5. Conclusion

We identified 12 members of the HDAC family in BPH. Sequence analysis revealed that the classic HDAC family of zinc-dependent deacetylases (NlHDAC) and the sirtuin family of $NAD^+$-dependent deacetylases (NlSIRT) contained six members each. Fecundity assay showed that RNA-mediated knockdown of *NlHdac1*, *NlHdac3* and *NlHdac4* but not other members played essential roles in female fertility through regulating ovary maturation or ovipositor development. The NlHDAC1 is the main HDAC in ovaries of BPH, and RNA-seq analysis showed that it regulated ovary maturation by multiple signalling pathways. The BPH males with *NlHdac1* knockdown lost the ability to make courtship songs, which impaired their mating success and led to male infertility.

Data accessibility. The material and datasets used and/or analysed during the current study are available from the corresponding author on reasonable request. The raw data from the RNA-seq were submitted to GenBank (SRA accession number: SRP152996). DNA sequences in Genbank were as follows: *NlHdac1* (LOC111055806), *NlHdac3* (LOC111062027), *NlHdac4* (LOC111044755), *NlHdac6* (LOC111050247), *NlHdac11* (LOC111043560), *NlHdac11_l* (LOC111045791), *NlSirt1* (LOC111062070), *NlSirt2* (LOC111044349), *NlSirt3* (LOC111063696), *NlSirt4* (LOC111056564), *NlSirt6* (LOC111058191) and *NlSirt7* (LOC111058155).

Authors' contributions. J.-L.Z. and H.-J.X. designed the experiment, performed data analysis and wrote the paper. X.-B.Y., S.-J.C., H.-H.C., N.X., W.-H.X., S.-J.F. and C.-X.Z. helped perform experiments and data discussion. H.-J.X. managed and directed the project.

Competing interests. The authors declare that they have no competing interest.

Funding. This work was supported by the National Science Fund for Excellent Young Scholars (31522047), the National Natural Science Foundation of China (31772158), the Zhejiang Provincial Natural Science Foundation for Distinguished Young Scholars (LR16C140001) and the Fundamental Research Funds for the Central Universities (2015XZZX004-34).

Acknowledgements. We thank Miss Dan-Ting Li for preparing figure 10.

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
