## [Reviewer comments · Open Biology]

Review History

RSOB-18-0158.R0 (Original submission)

Review form: Reviewer 1

Recommendation

Accept with minor revision (please list in comments)

Are each of the following suitable for general readers?

- a) **Title**
Yes

- b) **Summary**
Yes

- c) **Introduction**
Yes

Is the length of the paper justified?

Yes

Should the paper be seen by a specialist statistical reviewer?

No

Is it clear how to make all supporting data available?

Yes

Is the supplementary material necessary; and if so is it adequate and clear?

Yes

Do you have any ethical concerns with this paper?

No

Comments to the Author

This is a fairly straight forward research. Although the idea is not completely novel, the testing is exhaustive and robust. The methodology, for the most part, is clear. The flow and organization of the manuscript are acceptable, albeit it needs to be streamlined. Also, authors could improve the phenotypic impact part of the research by quantifying the mating behavior with statistical power to go with the existing schematic drawings (Figure 10) and the acoustic/auditory profiles (Figure 11). Finally, this manuscript will benefit from more thorough English editing. The detailed comments and suggestions to improve the manuscript are provided in the attached PDF.

Review form: Reviewer 2

Recommendation

Accept with minor revision (please list in comments)

Are each of the following suitable for general readers?

- a) **Title**
Yes
- b) **Summary**
Yes
- c) **Introduction**
Yes

Is the length of the paper justified?

Yes

Should the paper be seen by a specialist statistical reviewer?

No

Is it clear how to make all supporting data available?

Yes

Is the supplementary material necessary; and if so is it adequate and clear?

Yes

Do you have any ethical concerns with this paper?

No

Comments to the Author

Zhang and coworkers identified several genes encoding histone deacetylase in *Nilaparvata lugens*. They study the function of NIHDAC1, NIHDAC3, and NIHDAC4, which are involved in female fertility. They show that NIHDAC1 is likely the main histone deacetylase in ovaries. They demonstrate the function of NIHDAC1 using a variety of approaches, from RNA-seq analysis to behavioral studies. Their investigation has been thoroughly performed and describe in detail the different phenotypes of NIHDAC1. It is an excellent manuscript exhaustively covering a phenotype with potential for pest control.

The only criticisms I would raise is that the results show the effect of a pleiotropic gene, as expected by an overall regulator of chromatin function. Therefore, it is difficult to claim that a particular pathway is affected when the RNAseq experiments shows an effect in thousand of genes. I would suggest the authors to consider this point in the discussion, rather than listing all pathways possibly involved.

Decision letter (RSOB-18-0158.R0)

29-Oct-2018

Dear Dr Xu

We are pleased to inform you that your manuscript RSOB-18-0158 entitled "The histone deacetylase NIHDAC1 regulates both female and male fertility in the brown planthopper, *Nilaparvata lugens*" has been accepted by the Editor for publication in *Open Biology*. The reviewer(s) have recommended publication, but also suggest some minor revisions to your manuscript. Therefore, we invite you to respond to the reviewer(s)' comments and revise your manuscript.

Please submit the revised version of your manuscript within 14 days. If you do not think you will be able to meet this date please let us know immediately and we can extend this deadline for you.

When submitting your revised manuscript, you will be able to respond to the comments made by the referee(s) and upload a file "Response to Referees" in "Section 6 - File Upload". You can use this to document any changes you make to the original manuscript. In order to expedite the

processing of the revised manuscript, please be as specific as possible in your response to the referee(s).

- 1) A text file of the manuscript (doc, txt, rtf or tex), including the references, tables (including captions) and figure captions. Please remove any tracked changes from the text before submission. PDF files are not an accepted format for the "Main Document".
- 2) A separate electronic file of each figure (tiff, EPS or print-quality PDF preferred). The format should be produced directly from original creation package, or original software format. Please note that PowerPoint files are not accepted.
- 3) Electronic supplementary material: this should be contained in a separate file from the main text and meet our ESM criteria (see <http://royalsocietypublishing.org/instructions-authors#question5>). All supplementary materials accompanying an accepted article will be treated as in their final form. They will be published alongside the paper on the journal website and posted on the online figshare repository. Files on figshare will be made available approximately one week before the accompanying article so that the supplementary material can be attributed a unique DOI.

Online supplementary material will also carry the title and description provided during submission, so please ensure these are accurate and informative. Note that the Royal Society will not edit or typeset supplementary material and it will be hosted as provided. Please ensure that the supplementary material includes the paper details (authors, title, journal name, article DOI). Your article DOI will be 10.1098/rsob.2016[*last 4 digits of e.g. 10.1098/rsob.20160049*].

- 4) A media summary: a short non-technical summary (up to 100 words) of the key findings/importance of your manuscript. Please try to write in simple English, avoid jargon, explain the importance of the topic, outline the main implications and describe why this topic is newsworthy.

Images

Data-Sharing

It is a condition of publication that data supporting your paper are made available. Data should be made available either in the electronic supplementary material or through an appropriate repository. Details of how to access data should be included in your paper. Please see <http://royalsocietypublishing.org/site/authors/policy.xhtml#question6> for more details.

Data accessibility section

Sincerely,

The Open Biology Team
mailto:openbiology@royalsociety.org

Reviewer(s)' Comments to Author:

Referee: 1

Comments to the Author(s)

This is a fairly straight forward research. Although the idea is not completely novel, the testing is exhaustive and robust. The methodology, for the most part, is clear. The flow and organization of the manuscript are acceptable, albeit it needs to be streamlined. Also, authors could improve the phenotypic impact part of the research by quantifying the mating behavior with statistical power to go with the existing schematic drawings (Figure 10) and the acoustic/auditory profiles (Figure 11). Finally, this manuscript will benefit from more thorough English editing. The detailed comments and suggestions to improve the manuscript are provided in the attached PDF.

Referee: 2

Comments to the Author(s)

Zhang and coworkers identified several genes encoding histone deacetylase in *Nilaparvata lugens*. They study the function of NIHDAC1, NIHDAC3, and NIHDAC4, which are involved in female fertility. They show that NIHDAC1 is likely the main histone deacetylase in ovaries. They demonstrate the function of NIHDAC1 using a variety of approaches, from RNA-seq analysis to behavioral studies. Their investigation has been thoroughly performed and describe in detail the different phenotypes of NIHDAC1. It is an excellent manuscript exhaustively covering a phenotype with potential for pest control.

The only criticisms I would raise is that the results show the effect of a pleiotropic gene, as expected by an overall regulator of chromatin function. Therefore, it is difficult to claim that a particular pathway is affected when the RNAseq experiments shows an effect in thousand of genes. I would suggest the authors to consider this point in the discussion, rather than listing all pathways possibly involved.

Author's Response to Decision Letter for (RSOB-18-0158.R0)

See Appendix A.

Decision letter (RSOB-18-0158.R1)

09-Nov-2018

Dear Dr Xu

We are pleased to inform you that your manuscript entitled "The histone deacetylase NIHDAC1 regulates both female and male fertility in the brown planthopper, *Nilaparvata lugens*" has been accepted by the Editor for publication in Open Biology.

Article processing charge

Please note that the article processing charge is immediately payable. A separate email will be sent out shortly to confirm the charge due. The preferred payment method is by credit card; however, other payment options are available.

Sincerely,

The Open Biology Team
mailto: openbiology@royalsociety.org

Appendix A

Thanks for the referees' helpful comments. We now respond to the referees' comments point to point as follows:

Referee: 1

Comments to the Author(s)

This is a fairly straight forward research. Although the idea is not completely novel, the testing is exhaustive and robust. The methodology, for the most part, is clear. The flow and organization of the manuscript are acceptable, albeit it needs to be streamlined. Also, authors could improve the phenotypic impact part of the research by quantifying the mating behavior with statistical power to go with the existing schematic drawings (Figure 10) and the acoustic/auditory profiles (Figure 11). Finally, this manuscript will benefit from more thorough English editing. The detailed comments and suggestions to improve the manuscript are provided in the attached PDF.

Response to Referee 1:

1. We quantified the courtship duration and copulation duration, which were shown in lines 415-418 and Figure S6, in the revised manuscript.
2. We did English editing as Referee 1 requested:
 - 2.1. In the "Abstract", we revised the words as the referee suggested.
 - 2.2. Lines 71-72, the sentence of "...hundreds of studies on HDAC1 function in cancer were reported..." was revised to be "...hundreds of studies on the growth-promoting activity of HDAC1 in human cancer were reported..."

- 2.3. Sentence in lines 82-84 have revised to be “Interestingly, a new biological function was assigned to Rpd3, which showed that wild type flies subjected to a seven hour training session formed a robust long-term courtship memory, but this phenotype was completely abolished in the *Rpd3* mutant [28].”, which are in lines 82-85 in the revised manuscript.
- 2.4. We revised the last paragraph of the introduction as the referee’s suggestion, which corresponds to lines 103-113 in the revised manuscript.
- 2.5. We added headings and subheadings in the “Material and methods” as the referee suggested. However, we are not sure if this fits the journal’s format.
- 2.6. We added headings and subheadings in the “results” part.
- 2.7. We added headings and subheadings in the “Discussion” part.
- 2.8. In the conclusion part, the referee suggested to add a schematic drawing to summarize the existing and the novel hypothesis examined in this study. We worry about a schematic drawing might make redundant since the outlined conclusion is easy enough to follow.

Referee: 2

Comments to the Author(s)

Zhang and coworkers identified several genes encoding histone deacetylase in *Nilaparvata lugens*. They study the function of NIHDAC1, NIHDAC3, and NIHDAC4, which are involved in female fertility. They show that NIHDAC1 is likely the main histone deacetylase in ovaries. They demonstrate the function of NIHDAC1 using a variety of approaches, from RNA-seq analysis to behavioral studies. Their investigation has been thoroughly

performed and describe in detail the different phenotypes of NIHDAC1. It is an excellent manuscript exhaustively covering a phenotype with potential for pest control.

The only criticisms I would raise is that the results show the effect of a pleiotropic gene, as expected by an overall regulator of chromatin function. Therefore, it is difficult to claim that a particular pathway is affected when the RNAseq experiments shows an effect in thousand of genes. I would suggest the authors to consider this point in the discussion, rather than listing all pathways possibly involved.

Response to Referee 2:

We agree with Referee 2 that HDACs serve as overall regulators of chromatin modification, thus producing pleiotropic effects. We have presented this point in the discussion part of the revised manuscript (lines 513-515).